# Do Social Ties Moderate the Association between Childhood Maltreatment and Gratitude in Older Adults? Results from the NEIGE Study

**DOI:** 10.3390/ijerph182111082

**Published:** 2021-10-21

**Authors:** Satomi Doi, Yuna Koyama, Yukako Tani, Hiroshi Murayama, Shigeru Inoue, Takeo Fujiwara, Yugo Shobugawa

**Affiliations:** 1Department of Global Health Promotion, Tokyo Medical and Dental University (TMDU), Tokyo 113-8510, Japan; 140362ms@tmd.ac.jp (Y.K.); tani.hlth@tmd.ac.jp (Y.T.); fujiwara.hlth@tmd.ac.jp (T.F.); 2Research Fellow of Japan Society for the Promotion of Science, Tokyo 102-0083, Japan; 3Research Team for Social Participation and Community Health, Tokyo Metropolitan Institute of Gerontology, Tokyo 173-0015, Japan; murayama@tmig.or.jp; 4Department of Preventive Medicine and Public Health, Tokyo Medical University, Tokyo 160-8402, Japan; inoue@tokyo-med.ac.jp; 5Division of International Medicine, Niigata University Graduate School of Medical and Dental Sciences, Niigata 951-8510, Japan; yugo@med.niigata-u.ac.jp

**Keywords:** gratitude, childhood maltreatment, emotional neglect, social tie, older population

## Abstract

Background: Childhood maltreatment can impede gratitude, yet little is known about the older population and its moderators. The aim of this study is to clarify the association between childhood maltreatment and levels of gratitude of the older population, and the moderating effect of social ties on the association. Methods: We analyzed the data of 524 community-dwelling older adults aged 65–84 years without functional disabilities in Tokamachi City, Niigata, Japan, collected for the Neuron to Environmental Impact across Generations (NEIGE) study in 2017. Using a questionnaire, the participants rated three types of childhood maltreatment before the age of 18 (physical abuse, emotional neglect, and psychological abuse), level of gratitude, and social ties. Results: We found an inverse association between emotional neglect and gratitude. Furthermore, emotional neglect was inversely associated with gratitude only for those with lower levels of social ties. Conclusions: Promoting social ties may mitigate the adverse impact of emotional neglect on the level of gratitude.

## 1. Introduction

Gratitude is defined as a life orientation toward the appreciation of others and life circumstances [1], and as a positive emotion experienced when people receive kindness from others [2]. Previous systematic reviews and meta-analyses of gratitude interventions have confirmed the effect of gratitude on one’s well-being [3], including greater happiness [1,4], reduction in depressive and anxiety symptoms [5], and improvement of physical health such as subjective sleep quality [6]. Therefore, gratitude is one of the important determinants of health.

Gratitude plays a more critical role in older age given that the level of gratitude becomes higher as one grows older [7]. Moreover, gratitude can be expected to work as a potential factor that alleviates health problems among older adults: the number of people who suffer from chronic diseases that lead to increased mental health disorders and impaired well-being increases with age [8,9], partially due to retirement and loss of loved ones [10,11,12]. Previous studies have found that older adults with higher levels of gratitude are more likely to show the advantages of physical and mental health [7,13,14].

Although the importance of gratitude, especially among the older population, has been much acknowledged, the mechanism which determines levels of gratitude has not yet been established. A previous study reported that some personality traits such as more agreeableness and less narcissism were correlated with higher levels of gratitude [15]. However, given the current argument that personality is the set of traits and styles that one exhibits and cannot be separated from and be in a causal relationship to behavior and emotion [16], the association between personality and gratitude does not give any clues to the underlying mechanisms that determine the level of gratitude. Moreover, as personality is not modifiable, modifiable factors need to be identified in order to determine the level of gratitude.

Childhood maltreatment including physical, emotional, and sexual abuse, along with physical and emotional neglect, may be possible determinants. Studies have shown the mediation of gratitude on the association between childhood maltreatment and health problems. For example, a previous study involving 358 Chinese college students showed that levels of gratitude moderated the association between emotional and physical neglect and sexual abuse with depressive symptoms [17]. In another study of 795 undergraduate women in the United States, the association between early family adversity (e.g., child maltreatment and exposure to parental substance abuse) and poor well-being (e.g., perceived stress and alcohol consumption) was mediated by lower levels of gratitude [18]. These previous studies only examined a young to middle adulthood population and did not take into account potential confounders such as childhood socioeconomic status and current depressive symptoms, which limit inference of causality of childhood maltreatment among older adults.

We hypothesized that childhood maltreatment decreases the level of gratitude among older adults and that the association may be moderated by social ties. Childhood maltreatment does not always result in poor health consequences. A nationally representative study of adults in Canada showed that more than half of participants with a history of childhood abuse reported good mental health and supportive relationships with family and friends, which may be important in increasing the likelihood of better mental health outcomes following child abuse [19]. A population-based study of Japanese older adults showed that adverse childhood experiences including maltreatment were associated with increased dementia incidence only for those with low social capital [20]. In the literature on gratitude, a previous study found a correlation between higher levels of perceived emotional social support from family and acquaintances and levels of gratitude in Japanese young women [21]. Therefore, we can assume that higher levels of social ties mitigate the association between childhood maltreatment and lower levels of gratitude.

This study aims to examine the association of childhood maltreatment and levels of gratitude, adjusting for possible confounders. Whether social ties moderate the association in older adults will be examined using data from the Neuron to Environmental Impact across Generations (NEIGE), which was established to investigate the social determinants of health among older people in rural areas in Japan.

## 2. Materials and Methods

### 2.1. Participants

We used data from the Neuron to Environmental Impact across Generations (NEIGE). Details of this study design have been described elsewhere [22]. Briefly, we conducted a baseline survey of community-dwelling older adults aged 65–84 years old without functional disabilities, defined as not being certified as eligible for long-term care insurance (LTCI) benefits [23], in Tokamachi City, Niigata Prefecture, Japan, in 2017. We randomly selected study participants (*n* = 1346) from four stratified groups by age (65–74 and 75–84 years) and residential area (Tokamachi center (downtown) and Matsunoyama (mountains)) based on the resident register and sent invitations to participate in the study via mail. A total of 527 people accepted the procedure of examination and participated in the study (participation rate: 39.2%). The analytic sample for the present study comprised 524 participants after excluding the participants with missing responses to the questions related to childhood maltreatment (*n* = 3). Participants were informed that participation in the study was voluntary and written informed consent was obtained from all study participants.

### 2.2. Measurements

*Gratitude.* Levels of gratitude were assessed using two items: “I have so much in life to be thankful for” and “I am grateful to a wide variety of people”. These items were selected from the Gratitude Questionnaire—6 [2] and used in a previous study [24]. Participants were asked to evaluate their overall life gratitude on a seven-point scale ranging from 1 (strongly disagree) to 7 (strongly agree). The correlation between the two items was r = 0.82. The mean of the two items was calculated for each participant to reflect their overall life gratitude; a higher score meant that the respondent had a higher level of gratitude. This gratitude score was observed to have convergent and discriminant construct validity as it is positively correlated with measures of adaptive psychological traits such as life satisfaction and sympathy, and negatively correlated with measures of maladaptive psychological traits such as loneliness and perceived stress among Japanese adults [24].

*Childhood maltreatment.* Childhood maltreatment was assessed using a self-reported questionnaire. The questionnaire consists of three questions, asking whether participants had the following experiences before the age of 18: physical abuse (“I was beaten by my parents and injured” with the response “yes”), emotional neglect (“I felt loved by my parents” with the response “no”), and psychological abuse (“I was hurt by my parents’ harsh words or insulted by my parents” with the response “yes”). Physical abuse was defined as not only being beaten but more severely injured, because the Japanese older population had been subjected to corporal punishment, or *taibatsu*, as discipline from their parents when they were children [25]. These experiences were extracted from the Adverse Childhood Experiences Study [26] by experts of Japanese childhood adverse experiences (content validity). As psychometric properties, the presence of each experience has been reported to increase the probability of other experiences in older Japanese adults [20]. These experiences have also been observed to have predictive validity, as they were associated with health problems among Japanese adults [20,27].

*Social ties*. Current social ties were assessed using questions on neighborhood ties and frequency of meeting friends [28,29]. Participants rated levels of neighborhood ties on a scale of 1 = “I have neighbors who help each other such as consulting with and borrowing daily necessities from each other”, 2 = “I have neighbors who I can stand talking to on a daily basis”, 3 = “I have a minimal relationship with neighbors limited to such things as saying hello”, and 4 = “I have no relationship with neighbors”, in which the responses were dichotomized with “1” equating to “high neighborhood ties”, and “2”, “3”, and “4” equating to “low neighborhood ties”. Participants also rated their frequency of meeting friends on a scale of 1 (four or more times per week) to 6 (never), in which the responses were categorized into three groups: more than once a week (high frequency), 1–3 times a month (middle frequency), and rarely (low frequency).

*Covariates*. Covariates were assessed by a self-administered questionnaire, as shown in Table 1. We included childhood economic hardship and years of education as other childhood environments. Childhood economic hardship was assessed by a question on whether the financial condition was difficult during childhood (yes/no response) [30]. Education was categorized into three groups by years of schooling (≤9, 10–12, or ≥13 years) [31]. We included adult socio-demographics (current annual standardized household income, longest-held occupation, and marital status). Longest held occupation was categorized as non-manual (professional, technical, or managerial workers), manual (clerical, sales/service, skilled/labor, or agricultural/forestry/fishery workers, or other), and no occupation [31]. Current depressive symptoms were assessed by the 15-item short form of the Geriatric Depression Scale (GDS) (Japanese version) [32].

### 2.3. Ethics

This study was conducted according to the guidelines of the Declaration of Helsinki and approved by the Ethics Committee of Niigata University (approval number: 2666).

### 2.4. Statistical Analysis

First, a multivariate linear regression model was built to examine the association between three types of childhood maltreatment (i.e., physical abuse, emotional neglect, and psychological abuse) and levels of gratitude. After estimating a crude model, Model 1 was adjusted for age, sex, and other childhood environments (economic hardship and years of education), which are theoretically considered as confounders based on the previous studies [30,33]. Model 2 was additionally adjusted for current depressive symptoms, which are associated with and closer to level of gratitude. Model 3 was further adjusted for potential mediators, which are adulthood socio-demographics (annual standardized income, longest-held occupation, and marital status). Models 2 and 3 were performed when the estimation in Model 1 was significant. A correlation matrix among variables is shown in Appendix A.

Second, after conducting multivariate linear regression without the interaction term (Model 1), we examined the moderation effects of neighborhood ties and frequency of meeting friends by multivariate linear regression including the interaction term (Model 2). We also used models to explore the association between childhood maltreatment and levels of gratitude stratified by levels of neighborhood ties (i.e., high and low), and by frequency of meeting friends (i.e., low, middle, and high), adjusted for age, sex, and other childhood environment and current depressive symptoms. In order to avoid the reporting bias, current depressive symptoms were adjusted in this model. As for adulthood socio-demographics, we did not adjust because of overadjustment.

All analyses were conducted using Stata, Version 15 with the significance level set at 0.05.

## 3. Results

### 3.1. The Distribution of Characteristics

The participants’ characteristics are summarized in Table 1. A total of 53.1% of the participants were female and the mean age was 73.5 (SD = 5.6) years old. The prevalence of those who experienced physical abuse, emotional neglect, and psychological abuse in childhood were 1.0% (*n* = 5), 11.3% (*n* = 59), and 4.0% (*n* = 21), respectively. A total of 42.7% of the participants reported that they had economic hardship during childhood, and 38.2% of the participants had been in school for less than 9 years. Severe depressive symptoms (GDS total score: ≥ 10) were experienced among 4.6% of the participants. The mean gratitude score was 6.29 (SD = 0.93), which was higher among females than males (6.35 (SD = 0.91) vs. 6.22 (SD = 0.95), respectively) (Appendix A).

### 3.2. The Association between Childhood Maltreatment and Levels of Gratitude

Table 2 shows the association between each type of childhood maltreatment and levels of gratitude. We found that participants with emotional neglect showed lower levels of gratitude (coefficient = −0.27, 95% CI = −0.52 to −0.01), whereas the coefficients of physical and psychological abuse were not significant. The coefficient of emotional neglect remains significant in Model 3, adjusted for age, sex, other childhood environments, current depressive symptoms, and adulthood socio-demographics (coefficient = −0.28, 95% CI = −0.53 to −0.03).

### 3.3. The Moderation Effect of Social Ties on the Association between Emotional Neglect and Gratitude

As for the moderation effect of neighborhood ties, the interaction between emotional neglect and neighborhood ties was significant (coefficient = 0.80, 95% CI = 0.30 to 1.31; *p* for interaction = 0.002) (Table 3). Among the participants with high neighborhood ties, emotional neglect was not associated with levels of gratitude (coefficient = 0.22, 95% CI = −0.13 to 0.57). In contrast, emotional neglect was associated with levels of gratitude among those with low neighborhood ties (coefficient = −0.57, 95% CI = −0.91 to −0.23) (Figure 1A and Appendix A).

As for the moderation effect of frequency of meeting friends (Table 4), the interaction between emotional neglect and high frequency of meeting friends was significant (high frequency: coefficient = 0.84, 95% CI = 0.16 to 1.53; *p* for interaction = 0.016). Among the participants who met friends more than once a month, emotional neglect was not associated with levels of gratitude (high frequency: coefficient = −0.08, 95% CI = −0.38 to 0.23; middle frequency: coefficient = −0.28, 95% CI = −0.80 to 0.24). In contrast, emotional neglect was associated with levels of gratitude among those who met friends rarely (low frequency: coefficient = −1.07, 95% CI = −1.87 to −0.27) (Figure 1B and Appendix A).

## 4. Discussion

We found that older adults with emotional neglect in their childhood showed a lower level of gratitude whereas other types of childhood maltreatment (i.e., physical and psychological abuse) were not associated with gratitude. Furthermore, the moderation effect of social ties on the association between emotional neglect and gratitude was observed. That is, older adults with higher levels of social ties showed no association between childhood emotional neglect and level of gratitude.

Our findings are consistent with a previous study using a younger adult sample. Wu et al. [17] found a strong and inverse association between childhood emotional neglect and a level of gratitude (r = −0.61, *p* < 0.001), which was the strongest association compared to other types of childhood maltreatment (i.e., emotional abuse, physical abuse, physical neglect, and sexual abuse) among Chinese college students. On the other hand, Wu et al. [17] also showed a link between emotional abuse, physical abuse, physical neglect, and sexual abuse and a low level of gratitude, which was not true in this study. This discrepancy may lie in the population difference in terms of the prevalence of childhood abuse. For example, Wu et al. used a sample with the prevalence of physical abuse was 14.8% [17], while in the current study, only 1% of participants reported to have experienced physical abuse. The prevalence of physical abuse in our study might be caused by the definition of physical abuse, which was defined as not only being beaten but more severely injured based on the Japanese context [25]. Consistent with our results, another previous study showed the prevalence of physical assault which assessed as one of ACEs in older adults was 2.7% [34]. In terms of psychological abuse, the prevalence (i.e., 4.0%) was lower compared with other studies in older adults (e.g., 9.6% in the United States [35] and 10.9% in Malaysia [36]). This lower prevalence in our study might be caused by the wording of the question. Due to the difference of the prevalence of childhood abuse, we might not have enough power to assess the association between childhood physical and psychological abuse. Additionally, the survival effect might explain the lack of association of childhood physical and psychological abuse with gratitude. That is, older adults with childhood experiences of physical and psychological abuse might be more likely to have functional disabilities [37,38] (i.e., exclusion criteria in our survey) or more likely to die [39,40]. Nonetheless, the participants who reported the experience of physical abuse and psychological abuse in childhood and were only five and 21, respectively, which might cause insufficient statistical power. As for levels of gratitude, the distribution of our sample was skewed to the left (leaning towards higher scores), which is consistent with the previous study showing that in general, gratitude increases as one grows older [7].

We found that older people with experience of emotional neglect showed lower gratitude, which continues in later life, even after controlling for other childhood environment, adulthood socio-demographic status, and current depressive symptoms. Emotional neglect can be defined as a relationship pattern in which an individual’s affectional needs are consistently disregarded, ignored, invalidated, or unappreciated by a significant other [41]. Therefore, emotional neglect may be the essential risk factor for attachment disorders [42]. Moreover, emotional neglect self-reported in old age may reflect disorganized attachment, allowing us to speculate that attachment disorder may underlie the association between emotional neglect and lower gratitude. Considering that gratitude is a positive reaction to others when given personal benefits intentionally [43], which may be primarily and fundamentally experienced through the parent-child attachment process, the role of attachment in shaping later gratitude is reasonable. Previous studies reported that disorganized attachment with parents was related to fewer experiences of gratitude and lower gratitude among university students and young couples [43]. A recent longitudinal study also showed that more childhood attachment anxiety and avoidance predicted less adolescent gratitude [44].

Although older people with experience of emotional neglect showed lower gratitude, we also found that neighborhood ties and the frequency of meeting friends mitigated the association. These results were consistent with the previous studies which found that the presence of a warm and trusted partner and social support from family or friends modified the associations of history of childhood maltreatment with depression, general health, sleep problems, and antisocial behavior [45], and found that older adults who have adverse childhood experiences including childhood maltreatment, but middle to high levels of social capital, did not have a risk for dementia [20]. This may explain why strong neighborhood ties and a higher frequency of meeting friends may lead to more opportunities for feelings of gratitude. That is, even if older people experienced neglect from their parents in childhood, they can feel grateful if they can build good relationships with non-parents, such as neighbors and friends. Even though it is possible for older adults who tend to feel gratitude to be more likely to perceive strong neighborhood ties and meet their friends frequently (i.e., reverse causality), our findings provide worthiness in promoting social ties in order to increase gratitude. However, in general, people with adverse childhood experiences including maltreatment are less likely to have social connectedness [46]. Therefore, longitudinal studies are warranted to examine whether promoting social ties despite having maltreatment in childhood leads to increasing gratitude among older adults.

The possible mechanisms to explain the impacts of social ties and health outcomes have been shown by Kawachi and Berkman [47] using two theoretical models. The first model is the main effect model: social ties may directly induce positive psychological states (e.g., a sense of purpose and belongings), which may benefit mental health by increasing motivation for self-care, such as regular exercise. The second model is the stress-buffering model: social ties may mitigate the impacts of stressful event on mental health. According to these theoretical models, our findings may suggest that social ties act as preventive against the decrease in gratitude and have significant role in increasing gratitude by enhancing positive affective states. Additionally, the effects of a community program to increase social interaction for older adults in Japan, which is called “Kayoino-Ba”, on improving health outcomes have been established [48,49]. Another previous Japanese study (*n* = 2159) [50] indicates that participation in “Kayoino-Ba” increased social participation aside from “Kayoino-Ba”, and older adults who increased the frequency of social participation were more likely to have the health information and health awareness. The community program designed to increase social ties for older adults may mitigate the adverse impacts of childhood maltreatment on gratitude.

There are some limitations to our study. First, childhood maltreatment was assessed via a self-reported retrospective questionnaire, which may induce recall bias. However, the reported prevalence was similar to the numbers reported in other studies of older Japanese people [33,51]. Second, we could not assess other types of childhood maltreatment such as childhood sexual abuse. However, this enabled us to retain a higher response rate since, in Japan, the participants may be less likely to respond to a questionnaire that included a question on sexual abuse from parents [52]. Third, the participants who reported the experience of childhood maltreatment, especially physical abuse and psychological abuse, were only five in number, which might not be sufficient to detect the association between childhood maltreatment and gratitude. Further study with a larger sample size is needed. Fourth, the participation rate in the current study was low (39.2%). Although our sample was randomly chosen, limiting the participants to adults without functional disabilities may have led to a selection bias, slightly decreasing the chance of adults who were not healthy from being selected. Fifth, our study showed limited generalizability as our study site was a rural area in one prefecture (i.e., Niigata prefecture). Further studies which verify our results using samples living in other prefectures are needed. Finally, our study was a cross-sectional design, which hinders inferring causality. We could not deny the possible pathway from a higher level of gratitude to stronger social ties. A longitudinal study to elucidate the moderating effect of social ties on the levels of gratitude is needed.

## 5. Conclusions

We found that older adults with the experience of emotional neglect in their childhood were less likely to show gratitude. Importantly, our findings indicate that a higher level of neighborhood ties and a higher frequency of meeting friends mitigate the association between childhood emotional neglect and gratitude. According to our findings, interventions to promote neighborhood ties and frequency of meeting friends need to be considered to promote gratitude. Furthermore, since social ties can be established not only by an individual but also a population approach, municipalities and local communities should take active roles in promoting gratitude among their older populations.

## Figures and Tables

**Figure 1 ijerph-18-11082-f001:**
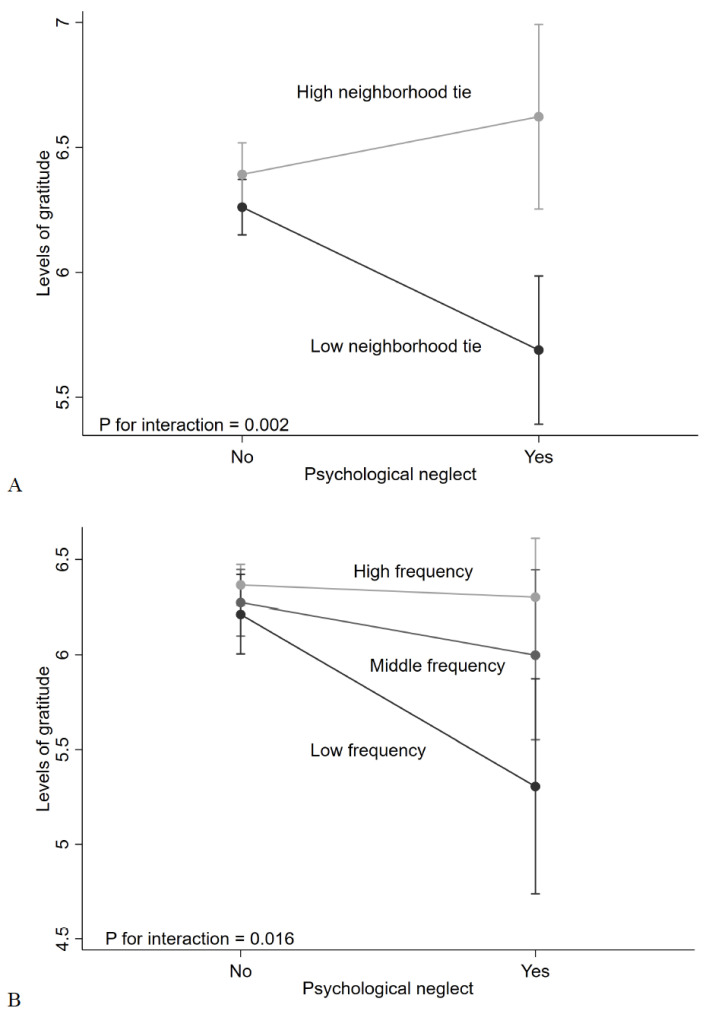
Association of emotional neglect with the levels of gratitude stratified by the levels of social ties: (**A**) the levels of neighborhood ties and (**B**) the frequency of meeting friends.

**Table 1 ijerph-18-11082-t001:** Characteristics of participants (*n* = 524).

	*n* or Mean	% or SD
Sex		
Male	246	46.9
Female	278	53.1
Age (years)	73.5	5.6
Childhood maltreatment		
Physical abuse	5	1.0
Emotional neglect	59	11.3
Psychological abuse	21	4.0
Childhood other environment		
Economic hardship		
No	300	57.3
Yes	224	42.7
Education (years)		
Low (≤9)	200	38.2
Middle (10–12)	218	41.6
High (≥13)	106	20.2
Current depressive symptoms (GDS)		
Non-depressed (<5)	407	77.7
Depressive symptoms (5–9)	89	17.0
Depressed (≥10)	24	4.6
Missing	4	0.8
Adult socio-demographics		
Annual income (million yen)		
Low (<2.00)	209	39.9
Middle (2.00–3.99)	230	43.9
High (≥4.00)	55	10.5
Missing	30	5.7
Longest occupation		
Non-manual	131	25.0
Manual	382	72.9
No occupation	10	1.9
Missing	1	0.2
Marital status		
Married	422	80.5
Widowed	83	15.8
Divorced/unmarried/other	11	2.1
Missing	8	1.5
Neighborhood tie		
High	226	43.1
Low	298	56.9
Frequency of meeting friends		
High (≥1/w)	312	59.5
Middle (1–3/m)	125	23.9
Low (Rarely)	87	16.6

GDS = Geriatric Depression Scale.

**Table 2 ijerph-18-11082-t002:** Associations between child maltreatment and gratitude among older Japanese adults (*n* = 524).

	Crude	Model 1	Model 2	Model 3
	Coefficient (95% CI)	Coefficient (95% CI)	Coefficient (95% CI)	Coefficient (95% CI)
Childhood maltreatment				
Physical abuse	0.21 (−0.61 to 1.04)	0.26 (−0.58 to 1.09)	-	-
Emotional neglect	**−0.28 (−0.54 to −0.03)**	**−0.27 (−0.52 to −0.01)**	**−0.27 (−0.51 to −0.02)**	**−0.28 (−0.53 to −0.03)**
Psychological abuse	0.02 (−0.39 to 0.43)	0.05 (−0.36 to 0.47)	-	-

CI = confidence interval. Model 1: Adjusted for age, sex, other childhood environment (economic hardship and education). Model 2: Model 1 + adjusted for current depressive symptoms. Model 3: Model 2 + adjusted for adult socio-demographics (annual income, longest occupation, and marital status). Note: Models 2 and 3 were performed when the coefficient was significant in Model 1. Bold signifies *p* < 0.05.

**Table 3 ijerph-18-11082-t003:** Associations of childhood emotional neglect and neighborhood ties with gratitude among Japanese older adults (*n* = 524).

	Model 1	Model 2
	Coefficient (95% CI)	Coefficient (95% CI)
Childhood maltreatment		
Emotional neglect	**−0.25 (−0.50 to −0.004)**	**−0.57 (−0.89 to −0.25)**
Neighborhood tie (ref = low)		
High	0.22 (0.06 to 0.39)	0.13 (−0.04 to 0.30)
Emotional neglect × Neighborhood tie		
Emotional neglect × High neighborhood tie		**0.80 (0.30 to 1.31)**

CI = confidence interval. Model: Adjusted for age, sex, other childhood environment (economic hardship and education), and current depressive symptoms. Bold signifies *p* < 0.05.

**Table 4 ijerph-18-11082-t004:** Associations of childhood emotional neglect and frequency of meeting friends with gratitude among Japanese older adults (*n* = 524).

	Model 1	Model 2
	Coefficient (95% CI)	Coefficient (95% CI)
Childhood maltreatment		
Emotional neglect	**−0.26 (−0.51 to −0.02)**	**−0.91 (−1.51 to −0.30)**
Frequency of meeting friends (ref = low)		
High	**0.25 (0.03 to 0.48)**	0.15 (−0.09 to 0.39)
Middle	0.14 (−0.12 to 0.39)	0.06 (−0.21 to 0.33)
Emotional neglect × Frequency of meeting friends		
Emotional neglect × High		**0.84 (0.16 to 1.53)**
Emotional neglect × Middle		0.63 (−0.14 to 1.41)

CI = confidence interval. Model: Adjusted for age, sex, other childhood environment (economic hardship and education), and current depressive symptoms. Bold signifies *p* < 0.05.

## Data Availability

All NEIGE datasets have ethical or legal restrictions for public deposition due to inclusion of sensitive information from the human participants. Data and analytic methods for the present study can be made available upon request.

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
