# Peer review of "Do Social Ties Moderate the Association between Childhood Maltreatment and Gratitude in Older Adults? Results from the NEIGE Study"

_ijerph, 2021, doi:10.3390/ijerph182111082_

Round 1

Reviewer 1 Report

Dear authors

Congratulations on the work you have been working on. Although I enjoyed reading your manuscript, I do believe your research presents important issues in terms of measurement that must be addressed in order to make this work suitable for publication.

Overall, the model that the author (s) sought to test is quite interesting. The selection of variables is wise and may be attractive for the scholar community; however, there are some issues that result concerning and lack of scientific rigor. 

My biggest concern is the validity of the instruments used to collect data. None of the scales report psychometric properties, a condition that may lead scholars to question the results of the study. Moreover, there are decisions that the author(s) made with respect to the use of certain items to measure certain constructs (e.g., gratitude). In this regard, it would be helpful for the reader to understand why the researchers made this decision; and overall, how they ensure the validity of this assessment. 

I also notice some issues related to the writing style (especially with tables and figures) and English grammar. So, it would be wise to get some help from a native speaker. 

The discussion section may be improved, overall including theoretical and practical implications.

Reviewer 2 Report

Manuscript ID: 1394269

A community-dwelling survey administered in Japan found a significant reverse association between one type of childhood maltreatment (psychological neglect) and gratitude among elderly people (aged 65-84 years old). After adjustment for a wide range of cofounders, the association remained significant. Social ties (measured as having frequent friends’ meetings and high neighborhood ties) moderated this association. Overall speaking, the contribution of the study is substantial, and the quality of the manuscript is acceptable. Following are the comments for the authors’ consideration:

Materials and Methods:

  • Emotional neglect sounds a better label than psychological neglect for the question on being (felt) loved by parents as a child. It helps to avoid confusion particularly given that you have included a question on “psychological abuse” too. It also is more consistent with the labeling used in the other ACEs/child maltreatment studies.

  • It is not clear whether the childhood maltreatment questionnaire is a standardized questionnaire or has been developed by the authors? If self-developed, have you made any attempt to measure its psychometric properties (e.g., convergent and/or divergent validity or other types of construct validity)? Compared with the items generally used in Adverse Childhood Experiences (ACEs) or child maltreatment studies, the question used in this study to measure physical abuse seems to be much stricter as beaten by parents during childhood was not enough to be considered as physical abuse, the child must have been injured too. If it was a matter of “or” and not “and”, more participants would probably answer affirmatively and you would get a higher prevalence rate (which means more statistical power for regression modeling). Compare this with the similar question used in the original ACEs study which is read as “Before age 18, did a parent or adult in your home ever hit, beat, kick, or physically hurt you in any way?  I believe this is one of the main reasons that the reported prevalence of this abuse in the current study is so low (only 1%,  5 participants). Internationally for this age bracket, a prevalence rate around10% or a higher has been reported. 

  • Similar comment as above applies to the question on Psychological abuse. The wording of the question might have contributed to the relatively low prevalence rate. A similar question used in the original ACEs study provide respondents with more options and covers less severe psychological abuse along with more severe one (it reads as “While you were growing up, in your first 18 years of life: Did a parent or adult in your home ever swear at you, insult you, or put you down?”)

  • Justify/explain why you have adjusted for these particular confounders? Any support from previous studies? Adding a correlation matrix can also help with this (see a comment on this below).

Results

  • I would suggest including Model 0 in Table 2 where no adjustment is made for any confounders.

  • To make sure the variables you included as confounders are truly acting as a confounder a correlation matrix is required between your main exposure (childhood maltreatment), sociodemographic characteristics during childhood and adulthood, and current depressive symptoms, and your main outcome of interest (gratitude).

  • Sub-heading 3.3. “The association between ACEs and time spent playing video games” is irrelevant!

  • Table 3 does not present the result on the moderation effect of the “Frequency of meeting friends” variable as stated in the manuscript, page 6, line 197. I wonder if Supplementary table 2 is presenting this moderation effect? If so, these are the main results and should be presented in the main table.

  • Make sure consistent labeling is used throughout the manuscript and in the Tables. Labels for categories/level of the Frequency of meeting friends variable in Table 1 are slightly different from those in Supplementary Table 2 (consistently use high, middle, low or ≥1/w, 1-3/m, rarely, choose one).

  • Half of the Results section is discussing results presented in the Supplementary Tables so I would not see them as supplementary but as the main results. I suggest moving them to the main text.

  • Dose β stand for standardized coefficients in the tables? If yes, clarify this. If that is the case, how would you interpret β > 1 for the low frequency of meeting friends presented in Supplementary table 3? If this is β and not B there is some serious multicollinearity between your predictors that you need to sort out.

  • Why models in Table 3 (and Supplementary Table 2) are not adjusted for adulthood socio-demographic characteristics? Explain this.

  • Have you conducted separate linear regression for each level of moderators in Supplementary table 3? If so, clarify this in the Method section.

Discussion

  • The non-significant association between physical abuse and gratitude seems to be due to floor effect than survival effect in this study as only one percent of the sample (5 people) responded affirmatively which I believe is a result of the wording you used to measure this variable. This is true that survival effect and/or selection bias (excluding those with a disability who are more likely to be subjected to childhood maltreatment) and/or recall bias all could potentially contribute to the low prevalence of reported physical abuse by the elderly (which has also been found in other international studies too), however in your study this is much less than expected. I am not surprised that no significant association was found between physical abuse and gratitude as you cannot expect to find a significant association only with a sample size of 5 people. Simply, your study does not have the statistical power to detect such an association and you need to be very clear on this limitation of your study in the discussion section.

  • The same comment as above applies to the "psychological abuse" variable and non-significant association with the outcome variable.

Minor comments

  • In the title, you could use “dose” instead of “how”. “Does Social Ties Moderate the Association Between Childhood Maltreatment and Gratitude in Older Adults? Results from the NEIGE Study”. I am suggesting this change because your analyses do not really show how this moderation happens. In the discussion section, you are proposing that disorganized attachment may explain “how” but you did not put this assumption to the test as this is not the focus of your study.

  • Make sure all people with initials in the Author's contribution are named as co-authors. In the current format MM, SS, GS are not named as a co-author on the first page (after the title).

  • This manuscript has an observational design, the word “experiment” mentioned in the Author's contribution is not appropriate to describe an observational study.

Round 2

Reviewer 1 Report

Dear authors, I appreciate the changes you made. They certainly improved the quality of your manuscript. I notice you address all my suggestions, you did a good job. 

Reviewer 2 Report

The authors have done a good job revising the manuscript and addressing my comments. No further comments from me.